# Perforated Appendicitis and Bowel Incarceration within Morgagni Hernia: A Case Report

**DOI:** 10.3390/medicina57020089

**Published:** 2021-01-21

**Authors:** Milica Mitrovic, Aleksandra Jankovic, Jelena Djokic Kovac, Ognjan Skrobic, Aleksandar Simic, Predrag Sabljak, Nenad Ivanovic

**Affiliations:** 1Center for Radiology and Magnetic Resonance Imaging, Clinical Centre of Serbia, Pasterova No. 2, 11000 Belgrade, Serbia; jankovicm.alex@yahoo.com (A.J.); jelenadjokic2003@yahoo.co.uk (J.D.K.); 2Department for Radiology, School of Medicine, University of Belgrade, Dr Subotica No. 8, 11000 Belgrade, Serbia; 3Hospital for Digestive Surgery, First Surgical University Hospital, Clinical Centre of Serbia, Koste Todorovica Street, No. 6, 11000 Belgrade, Serbia; skrobico@gmail.com (O.S.); apsimic65@gmail.com (A.S.); predragsabljak63@gmail.com (P.S.); nekic85@gmail.com (N.I.); 4Department for Surgery, School of Medicine, University of Belgrade, Dr Subotica No. 8, 11000 Belgrade, Serbia

**Keywords:** Morgagni hernia, perforated appendicitis, intestinal obstruction, emergency surgery

## Abstract

Morgagni hernia (MH) is a result of abdominal organ protrusion through the congenital defect in the anterior retrosternal aspect of the diaphragm. The colon and omentum are the most commonly involved organs, followed by the small intestine, stomach and liver. Symptoms of MH may be absent, although the majority of patients will experience mild dyspnea or abdominal discomfort. We present a case of MH complicated with intrathoracic acute perforated appendicitis and intestinal obstruction.

## 1. Introduction

Morgagni hernia (MH) develops when abdominal organs protrude through the congenital defect in the anterior retrosternal aspect of the diaphragm, which results from failure of fusion of the costal and sternal diaphragmatic attachments [1]. It is usually located on the right side, and rarely on the left side or bilaterally. The omentum and colon are the organs that most commonly herniate into the thorax, followed by the small intestine, and—rarely—the stomach or liver. Although it might be clinically silent, symptoms in adult populations range from pulmonary complaints, pain and discomfort in the thorax and abdomen. In rare cases it may present as an acute condition, most commonly due to intestinal obstruction [2].

Here, we represent a case of an unusual thoracoabdominal emergency, due to a perforated appendicitis within the Morgagni hernia, associated with intrathoracic bowel incarceration.

## 2. Case Report

A 40-year-old female was admitted to the emergency unit of our hospital with chest pains located on the right and irradiating in the right scapular region. She was febrile, fatigued, with complaints of mild bloating, and irregular stool in terms of diarrhea, and inability to pass gases. She did not complain of nausea or vomiting. The primary symptom was the sudden onset of chest pain that occurred three days prior to hospital admission, followed with a fever and mild bloating. Clinical examination revealed weakened respiratory function on the right side of the chest during auscultation, with normal breathing on the left. There was slight abdominal pain during palpation. Abdominal peristalsis was present, however it was dull and blunted. Laboratory examinations showed the presence of a strong inflammatory syndrome, with C-reactive protein levels of 327.2 mg/L, and a white blood cell count of 24,100/μL, with marked hyperneutrophilia. Chest X-ray showed air-filled bowel loops in the lower parts of the right hemithorax combined with pleural effusion. Abdominal X-ray revealed several hydroaeric levels, and no signs of bowel dilatation. Further, computed tomography (CT) of the chest and abdomen was performed. This, showed an anteriorly located diaphragmatic defect with herniation of omentum, right colon and small bowel loops in the right hemithorax (Figure 1). Within the hernia sac, signs of inflammation were obvious. Cecum and ileum were involved with extensive surrounding fat stranding, an abscess formation in the pericecal region and small volume of free fluid. Additionally, reactive right-sided pleural effusion was present (Figure 2). The diagnosis of a large Morgagni hernia with signs of inflammation within the sac was made. Although the appendix was not clearly seen, findings were highly suggestive for an acute appendicitis, complicated with abscess formation due to perforation.

Patient underwent surgery the following day after admission. At the time of diagnosis, we could not estimate intrathoracic adhesions and the presence of intestinal damage, but due to good previous experience with MH reparation we opted for a laparoscopic approach. Laparoscopy was initiated by introducing one 10 mm trocar on the midline, 5 cm above umbilicus and two 5 mm trocars in left and right subcostal region. There was an obvious protrusion of the transverse and right colon trough the anterior diaphragmatic opening, along with the omentum and small bowel. An enlarged liver was noticed, covering almost the entire upper abdomen. Reposition of the bowel to the abdomen was not possible due to strong adhesions located inside the chest. A conversion was made to open surgery. An upper midline laparotomy was made, and the abdominal retractor was set for better exposure. An opening of 15 cm wide was identified on the right anterior portion of the diaphragm, with the described parts the of colon and small intestines inside. Manual reposition of the organs was performed with great difficulty due to strong adhesions inside the right chest. During reposition, a large amount of serosal fluid was evacuated from the chest, as well as approximately 200 mL of pus which was located around the ileocolic region. After reposition, fibrin layers were removed from the right thorax, lavage was performed. The right lung inflated, and the thoracic drain was placed into position. The diaphragmatic opening was dissected and prepared for the suture (Figure 3). There were signs of necrosis of the terminal ileum and caecum, with thick fibrin layers. The appendix was completely gangrenous, presenting as 10 mm long necrotic tissue, covered with fibrin (Figure 4). In this situation, ileocecal resection was performed, followed with ileo-colic anastomosis in two layers, side to side fashion, using a continuous 3.0 absorbable monofilament suture. Direct suture of the diaphragmatic opening was made using a 2.0 non-absorbable polyfilament suture. The postoperative course was uneventful. The patient restored normal digestive function, and commenced oral intake on postoperative day 4. She was discharged from hospital 7 days after surgery in a stable condition with a continuous decline in the values of inflammatory parameters. The postoperative values of C-reactive protein were 30.1 mg/L, white blood cell 12,500/μL, while her neutrophil count was 11,000/μL. A regular chest X-ray was performed one year after surgery, showing a fully expanded right lung, and no sign of recurrent herniation.

## 3. Discussion

Morgagni hernias arise at the level of the sternocostal triangle, when there is a defect in the junction of the septum transversum of the diaphragm to the costal arches and the xiphoidal process of the sternum. It was named after Giovanni Battista Morgagni, a famous Italian anatomist and the father of modern pathologic anatomy, who first described the condition in 1761, after an autopsy on an Italian stonecutter [1]. The foramen of Morgagni is covered with parietal pleura on the proximal side, and peritoneum on the distal side, separated by a thin layer of fat tissue. MH accounts for up to 2–3% of all diaphragmatic hernias, and 27% of hernias of congenital origin [2,3]. There is a known link between patients with congenital anomalies, especially Down’s syndrome and this rare form of congenital diaphragmatic hernia [4]. MH can rarely be associated with paraesophageal hernia. Several cases of simultaneous MH and paraesophageal hernia have been reported [5]. Although it has been reported that these hernias are commonly asymptomatic, a study by Horton et al., who conducted a literature based search and identified 298 cases, found that around 28% of patients were actually symptom free [6]. The most common symptoms of MH were of pulmonary origin, such as dyspnea, cough and shortness of breath, and these presented in 36% of patients. In 20% of patients the main symptoms were chest and abdomen pains, followed by obstruction. Very rarely, MH may show symptoms that require urgent surgical care, and if so, it is most often related to intestinal obstruction [7]. There are several case reports in the previous literature describing acute appendicitis inside the Morgagni hernia [8,9,10]. To our knowledge, this is the first case report describing both obstruction and perforated appendicitis related to a Morgagni hernia. However, in both symptomatic and asymptomatic patients, a surgical approach is advised in order to avoid morbidity as a result of incarceration or ischemia of abdominal structures in the chest [5].

In the present case, the patient did not encounter any of the typical MH related symptoms prior to the onset of appendicitis. The thoracic pain was most likely a result of inflammation that involved the parietal pleura. We can assume that the intestinal obstruction was a result of the intestinal edema caused by inflammation and malposition of the organs in the hernia.

The initial diagnostic work-up included a chest X-ray and blood analysis. However, when there is clinical suspicion of intrathoracic herniation, computed tomography should be performed, as it provides anatomical information of different hernia types, hernia sac content, and also, very importantly, the evidence of possible complications [11,12].

The surgical method of MH treatment consists of organ reposition and suture of diaphragmatic defect. This can be achieved through a transabdominal or transthoracic approach. Young et al. compared their results of the elective repair of MH in regard to the open or laparoscopic approach [13]. There were no differences when complications rates were evaluated, while the operative time was statistically significantly shorter in the group of patients with laparotomy, in contrast to the thoracotomy and laparoscopy group. As expected, laparoscopy provided a shorter hospital stay and patients experienced less postoperative pain. In the current literature there is plenty of evidence about the safety of the laparoscopic approach. [14,15,16]. The defect is usually sutured directly and reinforced with mesh. However, direct suture without the usage of a mesh also provides satisfactory results for long term recurrence prevention. In the laparoscopy cases, however, the majority of authors use a mesh in order to reinforce the suture [17]. The key concern regarding the mesh utilization, is the possibility of erosion trough the surrounding tissues and visceral organs. Usage of a composite covered mesh should significantly decrease these risks. Reinforcement of the suture line with the falciform ligament has been proposed as a viable alternative to using a mesh [18].

In this case, laparoscopic reposition of the organs was not possible due to strong adhesions inside the chest. After an open approach and reposition, direct suture was possible. We avoided the mesh placement due to heavy inflammation and the possibility of postoperative infection.

## 4. Conclusions

Although rare, the Morgagni hernia must be taken into consideration as a possible cause of thoracoabdominal emergency, especially when intestinal obstruction symptoms follow the dyspnea or/and thoracic pain. Computed tomography represents the most valuable diagnostic method and it should be used prior to surgery. MH related emergencies warrant prompt and decisive surgical solutions.

## Figures and Tables

**Figure 1 medicina-57-00089-f001:**
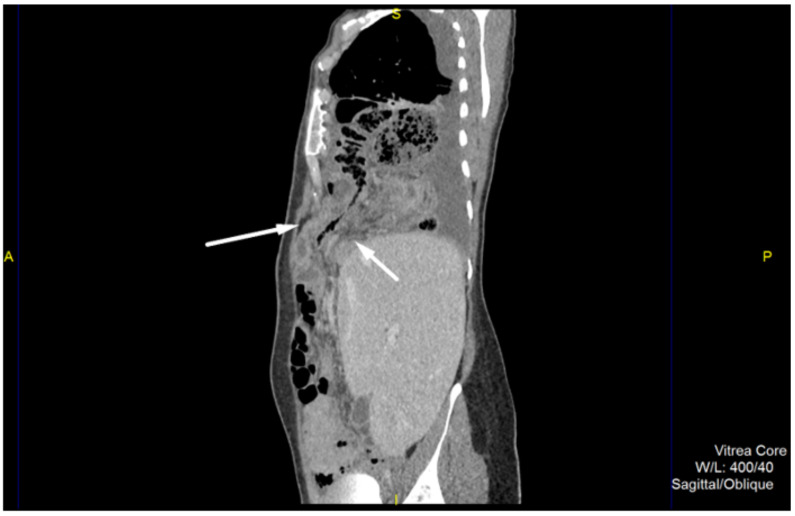
Contrast-enhanced computed tomography, sagittal view in portovenous phase shows an anteriorly located diaphragmatic defect with herniation of omentum, right colon and small bowel loops to the right hemithorax.

**Figure 2 medicina-57-00089-f002:**
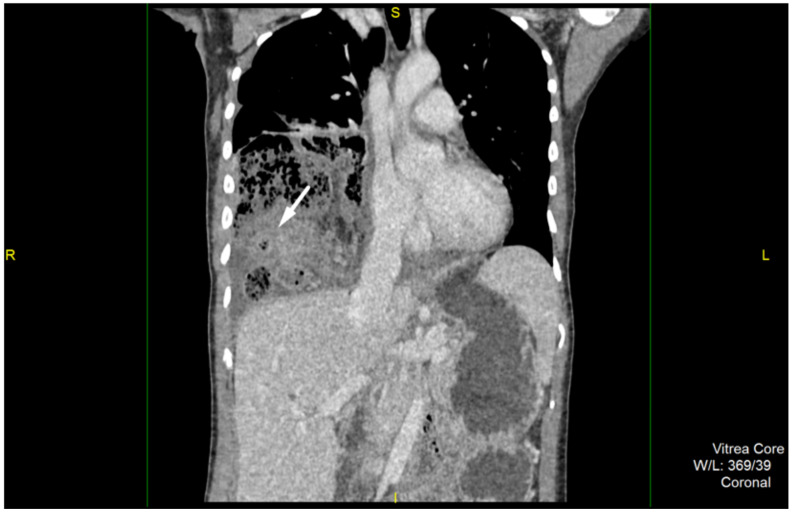
Contrast-enchanced computed tomography, coronal reconstruction reveals signs of inflammation involving cecum and ileum with extensive surrounding fat stranding.

**Figure 3 medicina-57-00089-f003:**
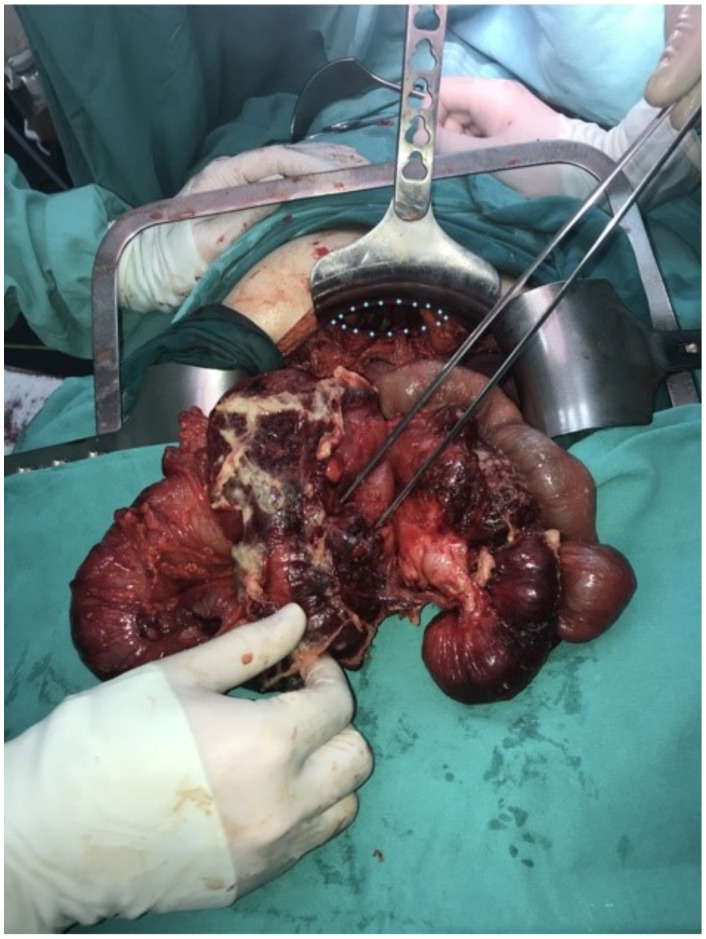
Necrotic terminal ileum and cecum covered with thick fibrin layers with perforated appendix and dilated small intestine after being repositioned into abdomen.

**Figure 4 medicina-57-00089-f004:**
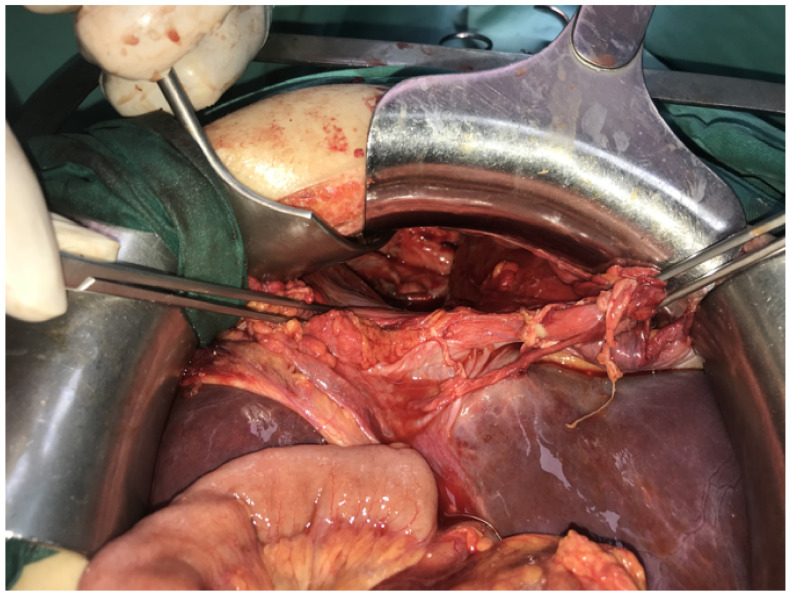
Morgagni hernia defect after the reposition of organs; intrathoracic visibly collapsed, right lower lung lobe with presence of fibrin and tissue debris.

## Data Availability

Article type is Case report. This statement is excluded.

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
