# Peer review of "Perforated Appendicitis and Bowel Incarceration within Morgagni Hernia: A Case Report"

_medicina, 2021, doi:10.3390/medicina57020089_

Round 1

Reviewer 1 Report

In this paper the authors present a case report of a patient with a rare complication of a Morgagni hernia.

In this paper the authors present a case report of a patient with a rare complication of a Morgagni hernia.

The case is well presented and the CT images are useful for understanding of the case. Some of the phrasing of the paper is unorthodox but sufficient for understanding.

The labels for the photos in figure 3 and figure 4 appear to be wrong – I think for figure 3 the label should be ‘Necrotic terminal ileum and cecum covered with thick fibrin layers with perforated appendix and dilated small intestine after being repositioned into abdomen’ for the photo showing the hernia it would be helpful for the authors to add a dotted line around the hernial orifice to the photo to help the reader.

It would be worthwhile for the authors to discuss why they initially chose a laparoscopic approach given the CT findings and signs of inflammation – was this approach initially chosen due to patient preference and did the surgeons feel it would be feasible for this hernia to be repaired laparoscopically?

The discussion could be developed a bit further with reference to left and right sided hernias and the association of Morgagni hernias in those with chromosomal abnormalities such as these papers have described:

  • Bettini A, Ulloa JG, Harris H. Appendicitis within Morgagni Hernia and simultaneous Paraesophageal Hernia. BMC Surg. 2015 Feb 2;15:15. doi: 10.1186/1471-2482-15-15. PMID: 25644716; PMCID: PMC4417238.
  • Intra-thoracic appendicitis in a child with Down’s syndrome, Parsons, Chris et al. Journal of Pediatric Surgery, Volume 48, Issue 6, e29 - e31

Author Response

Dear Reviewers ,

We would like to express our sincere appreciation for your valuable comments on our paper

“ Perforated appendicitis and bowel incarceration within Morgagni hernia: a case report” . We have given the comments serious consideration and altered the manuscript according to the suggestions.

We hope that the revised manuscript will meet your expectations and we are willing to consider all the further revisions. The revisions have been approved by all authors and the revised manuscript is attached. Thank you for your interest in our manuscript!     

Comments and answers:

Reviewer 1

  1. The labels for the photos in figure 3 and figure 4 appear to be wrong – I think for figure 3 the label should be ‘Necrotic terminal ileum and cecum covered with thick fibrin layers with perforated appendix and dilated small intestine after being repositioned into abdomen’ for the photo showing the hernia it would be helpful for the authors to add a dotted line around the hernial orifice to the photo to help the reader.

Authors: Corrections have been made in revised manuscript

A dotted line was added in Figure 3.

  1. It would be worthwhile for the authors to discuss why they initially chose a laparoscopic approach given the CT findings and signs of inflammation – was this approach initially chosen due to patient preference and did the surgeons feel it would be feasible for this hernia to be repaired laparoscopically?

Authors: We have chosen laparoscopic approach initialy, as we have had good previous experience with Morgagni hernia repair. Moreover at the time of diagnosis we couldn’t estimate intrathoracic adhesions and presence of intestinal damage.

Added in revised manuscript    Ln 59-61

  1. The discussion could be developed a bit further with reference to left and right sided hernias and the association of Morgagni hernias in those with chromosomal abnormalities such as these papers have described:
  • Bettini A, Ulloa JG, Harris H. Appendicitis within Morgagni Hernia and simultaneous Paraesophageal Hernia. BMC Surg. 2015 Feb 2;15:15. doi: 10.1186/1471-2482-15-15. PMID: 25644716; PMCID: PMC4417238.
  • Intra-thoracic appendicitis in a child with Down’s syndrome, Parsons, Chris et al. Journal of Pediatric Surgery, Volume 48, Issue 6, e29 - e31

Authors: Your comment is quite accurate and thank you for pointing us to it. We have improved our list of references with these papers and discussed them in the revised manuscript Ln 102-105

Ln 114-116

Ln 160-163

Reviewer 2 Report

Authors have done excellent work. Cases details ma be well explained

Author Response

Dear Reviewers,

We would like to express our sincere appreciation for your valuable comments on our paper

“ Perforated appendicitis and bowel incarceration within Morgagni hernia: a case report” . We have given the comments serious consideration and altered the manuscript according to the suggestions.

We hope that the revised manuscript will meet your expectations and we are willing to consider all the further revisions. The revisions have been approved by all authors and the revised manuscript is attached. Thank you for your interest in our manuscript!     

Comments and answers:

Reviewer 2

  1. Cases details ma be well explained.

Authors: Thank you for the positive comments, the manuscript has been improved with additional data in the case report section as well as in the discussion, along with two new references.

Reviewer 3 Report

The manuscript by Mitrovic et al presents a case study of 40 year old female who had thoraco-abdominal emergency due to perforated appendicitis within Morgagni hernia. The authors took open approach and performed manual reposition of organs. The patient was discharged 7 days from the surgery and the patient and started oral intake on day 4. Overall the manuscript looks very good and I have no major comments to the authors. Minor point - the authors could include the full blood profile and the X-ray from follow-up, latter not necessary.

Author Response

Dear Reviewers ,

We would like to express our sincere appreciation for your valuable comments on our paper

“ Perforated appendicitis and bowel incarceration within Morgagni hernia: a case report” . We have given the comments serious consideration and altered the manuscript according to the suggestions.

We hope that the revised manuscript will meet your expectations and we are willing to consider all the further revisions. The revisions have been approved by all authors and the revised manuscript is attached. Thank you for your interest in our manuscript!     

Comments and answers:

Reviewer 3

  1. Minor point - the authors could include the full blood profile and the X-ray from follow-up, latter not necessary.

Authors: Corrected in revised manuscript Ln 81-83

Thank you for the constructive advice and comments you sent us. We hope that the revised manuscript will receive an affirmative answer.